# Compensation of the Stress Gradient in Physical Vapor Deposited Al_1−x_Sc_x_N Films for Microelectromechanical Systems with Low Out-of-Plane Bending

**DOI:** 10.3390/mi13081169

**Published:** 2022-07-24

**Authors:** Rossiny Beaucejour, Michael D’Agati, Kritank Kalyan, Roy H. Olsson

**Affiliations:** 1Department of Mechanical Engineering and Applied Mechanics, University of Pennsylvania, 220 S. 33rd St., Philadelphia, PA 19104, USA; rossiny@seas.upenn.edu; 2Department of Electrical and Systems Engineering, University of Pennsylvania, 3205 Walnut St., Philadelphia, PA 19104, USA; mdagati@seas.upenn.edu; 3Singh Center for Nanotechnology, University of Pennsylvania, 3205 Walnut St., Philadelphia, PA 19104, USA; kritank@seas.upenn.edu

**Keywords:** aluminum scandium nitride, physical vapor deposition, stress, stress gradient, fabrication, cantilever beams, MEMS

## Abstract

Thin film through-thickness stress gradients produce out-of-plane bending in released microelectromechanical systems (MEMS) structures. We study the stress and stress gradient of Al_0.68_Sc_0.32_N thin films deposited directly on Si. We show that Al_0.68_Sc_0.32_N cantilever structures realized in films with low average film stress have significant out-of-plane bending when the Al_1−x_Sc_x_N material is deposited under constant sputtering conditions. We demonstrate a method where the total process gas flow is varied during the deposition to compensate for the native through-thickness stress gradient in sputtered Al_1−x_Sc_x_N thin films. This method is utilized to reduce the out-of-plane bending of 200 µm long, 500 nm thick Al_0.68_Sc_0.32_N MEMS cantilevers from greater than 128 µm to less than 3 µm.

## 1. Introduction

Microelectromechanical System (MEMS) structures are utilized to form cantilevers for sensors [1,2], resonators [3], and piezoelectric energy harvesters [4]. In piezoelectric energy harvesters [4] and micro-accelerometers [5], the properties of the cantilevers and their geometric limits directly impact device performance. The amount of deformation in cantilever beams and other structures depends on the growth technique utilized and material selection.

Aluminum Nitride (AlN) is a piezoelectric material commonly selected for the fabrication of vibrating MEMS structures. In 2009, it was determined that alloying AlN with scandium (Sc) can increase the piezoelectric coefficients by 500% [6]. AlN and Aluminum Scandium Nitride (Al_1−x_Sc_x_N) films deposited using Molecular Beam Epitaxy (MBE) or metal-organic Chemical Vapor Deposition (MOCVD) demonstrate exceptional film quality and high electromechanical coupling (k_t_^2^) but require processing at high temperature [7,8] and exhibit challenges due to stress gradients in the epi layers and complications in the releasing of structures at the end of the fabrication process [8]. For example, Park et al. [8] developed MBE Al_1−x_Sc_x_N Lamb wave resonators at super high frequencies directly on Silicon with acoustic velocities of 14,000 m/s and k_t_^2^ of 7.2%, but compressive film stresses produced buckling in the released films [9].

Sputtering is a Physical Vapor Deposition (PVD) method used to produce semi-conformal films of controlled thickness with high deposition rates at low substrate temperatures for realizing MEMS devices. During sputter deposition, ions accelerate and bombard a target removing atoms to be deposited on the wafer. There are numerous collisions that impact the mean free path of atoms and which control coverage on all surfaces. These processes also introduce stresses within the resulting film [10,11]. Reush et al. grew 001 polycrystalline AlN films using reactive RF magnetron sputtering directly on Silicon [7]. During growth, differences in grain size induced forces between grains at the grain boundaries. The increase in grain size with increasing thickness produced an intrinsic stress gradient through the thickness caused by zone T growth [12] and self-shadowing [6,13]. Knisely et al. determined that stress gradients in AlN film growth were a result of film growth stress, nucleation, and coalescence [14]. Knisely [14] further demonstrated a method to control both the stress and through-thickness stress gradient in AlN thin films using radio frequency (RF) substrate bias power modeled using a power law function. Introduction and further increases in RF substrate bias power during the sputtering of AlN based materials results in films with more compressive stress [14]. For Al_1−x_Sc_x_N films with high Sc substitution for Al, the resulting average film stress is often compressive [3,15] without the introduction of an RF substrate bias, especially when optimizing for low anomalously oriented grain (AOG) growth. Thus, it may not be possible to utilize RF substrate bias to simultaneously optimize for low average film stress and low through-thickness stress gradient in highly Sc alloyed AlN thin films.

Several stress and stress control techniques have been produced for MEMS devices. Pulskamp et al. mitigated the stress and stress gradient in sol-gel Lead Zirconium Titanate (PZT) multi-layer MEMS devices by altering the silane flow rates, annealing the top layer Platinum (Pt), and adjusting the film thickness [16]. For the 0.5 µm devices, a Titanium/Platinum 651 MPa (Ti/Pt) adhesive layer produced a stress of 35 MPa in the PZT with displacements of −250 µm [16]. Characterizing the released structures with analytical modeling and controlling process variability during growth further characterized the stress within these films [16]. Devices such as RF switches, accelerometers, and optical mirrors are very sensitive to the stress and stress gradient [9,17]. Sedky et al. developed these MEMS devices with p-type Silicon Germanium (Si_x_Ge_1−x_) using low pressure chemical vapor deposition (LPCVD) [17]. The stress was optimized by modifying the Se concentration from 40% to 90%, tuning the pressure from 650 mtorr to 800 mtorr, and controlling the deposition temperature to be between 400 and 450 °C. The stress gradient was reduced with a laser annealing technique which reduced stress from 125 MPa to 25 MPa and deflections from 118 µm to 5 µm [17]. Zhu determined the stress was caused by variations in film growth resulting in uneven grains and nonuniform boundaries [18]. When Aluminum Nitride (AlN) and Ruthenium (Ru) is sputtered to fabricate MEMS resonators, a folded beam design was used to limit deflection [18]. Mulloni et al. used similar clamped-clamped and single-clamped cantilever designs for Gold microstructure using electrodepositing [19]. The clamped-clamped released structures were more sensitive to stress variations than the single-clamped design [19]. Depending on the desired bending, the release temperature was reduced to limit stress which increased fabrication time [19]. These methods including temperature control, thickness variation, process control, laser annealing, design alterations, and pressure tuning show promise in reducing stress and stress gradients in other material systems.

Previous studies have explored methods using total N_2_ process gas flow to control average Al_1−x_Sc_x_N film stress [3,20,21]. These studies show that total process gas flow and the process gas mixture have a significant impact on the crystallinity, defects, and stress of sputter-deposited Al_1−x_Sc_x_N films [3,20,21]. Al_1−x_Sc_x_N films with a high degree of crystallinity and few defects require long mean free path and thus low process pressures. Sputtering Al_1−x_Sc_x_N at low process pressure in a pure N_2_ environment improves c-axis orientation and decreases surface roughness [3]. Increases in N_2_ flow increases ion scattering, reduces ion mean free path (MFP) and increases average film stress [20,21,22]. A method to suppress the stress gradients which produce out-of-plane bending while holding Sc concentration constant, maintaining film quality, and minimizing average stress has not been established for Al_1−x_Sc_x_N thin film materials. We previously demonstrated that 500 nm thick Al_0.68_Sc_0.32_N materials, free of AOGs, with low surface roughness and strong c-axis orientation, could be sputter-deposited on Si with a controlled average film stress ranging from −458 to 287 MPa, by controlling N_2_ flow between 20–30 sccm [3]. Here we provide a novel method that demonstrates both low average film stress and low through-thickness film stress gradient simultaneously by varying the N_2_ flow within the 20–30 sccm range during the deposition of Al_0.68_Sc_0.32_N. Furthermore, we demonstrate the utility of this method by fabricating Al_0.68_Sc_0.32_N MEMS cantilevers with low out-of-plane bending.

Al_1−x_Sc_x_N with high Sc alloying is a promising piezoelectric material for MEMS radio frequency (RF) filtering [3], energy harvesting [4], and sensing [23,24] devices because of its demonstrated high figure of merit [2] for each of these application areas. Piezoelectric RF bulk acoustic wave (BAW) resonator filters can often be implemented as anchored plates with extremely high out-of-plane bending stiffness. Such an implementation allows the realization of high-performance BAW resonators and filters using Al_1−x_Sc_x_N materials with high through-thickness stress gradient [3]. By contrast, high performance energy harvesters [4] and piezoelectric sensors [24,25] often require implementations using compliant cantilever [24] and clamped-guided beams [25] with low out-of-plane bending stiffness. In these devices, large film through-thickness stress gradients and the resulting out-of-plane bending can cause a significant degradation of the on-axis sensitivity and a corresponding increase in the cross-axis sensitivity. In addition, high out-of-plane bending complicates wafer level packaging (WLP) of MEMS devices because the packaging cavity must be able to accommodate the out-of-plane bending displacement. Therefore, if high performance Al_1−x_Sc_x_N materials are to be utilized in sensing and energy harvesting applications it is imperative to achieve low through-thickness stress gradient for the Al_1−x_Sc_x_N film and low out-of-plane bending for the MEMS structures implemented with the Al_1−x_Sc_x_N film.

## 2. Background and Theory

### 2.1. Average Film Stress Measurement

Once released from a substrate, as is common in MEMS processing, films will relax their built-in stress, which can lead to undesired deformation and/or buckling. Thus, knowing the stress in each layer of a MEMS process is vitally important. The average film stress is computed by measuring the wafer’s radius of curvature with a profilometer and subsequently using the Stoney equation [3,14,20,26,27]. To separate the various components leading to bending of the substrate, the radius of curvature is measured both before, R_0_, and after, R, the deposition of each film, allowing the built-in stress of each layer, T_f,BI_, [3,14,20,26,27] in a process to be isolated
(1)$$T_{f,\mathrm{BI}}=\frac{1}{6}\frac{Y_{s}}{\left(1-v_{s}\right)}\frac{t_{s}^{2}}{t_{f}}\left(\frac{1}{R}-\frac{1}{R_{0}}\right)$$
where $Y_{s}$, $v_{s}$ and $t_{s}$ are the Young’s Modulus, Poisson’s ratio, and thickness of the substrate respectively, and $t_{f}$ is the thickness of the film.

### 2.2. Relationship between the Total Process Gas Flow and the Resulting Film Stress

Knisely [14] introduced a model describing the power law relation between the RF substrate bias and the resulting AlN film stress. For a fixed Sc alloy this power law equation can be adapted to instead describe the total Al_1-x_Sc_x_N films stress $T_{\mathrm{ave}}$ as a function of N_2_ process gas flow
(2)$$T_{\mathrm{ave}}=\alpha (\beta (t_{f}{)}^{\gamma }+F_{N_{2}})$$
where α is determined from the slope of the stress versus N_2_ gas flow, $F_{N_{2}}$ is the N_2_ gas flow in the chamber, and β and γ are empirical fits based on the deposition parameters and environment. Similar to Knisely [14] we implement a film that utilizes multiple layers deposited under different sputtering gas conditions where the average film stress of each layer is used to compensate for the through-thickness stress gradient. Where Knisely [14] utilizes a different RF substrate bias to control the stress of each AlN layer, we utilize a different N_2_ flow to control the stress of each Al_0.68_Sc_0.32_N layer. The layer film stress, t_f_, based on the thickness and N_2_ flow of the layer is derived from integrating the average film stress [14] and is given by
(3)$$T_{f}(t,F_{N_{2}})=\alpha (\beta (1+\gamma )(t_{f}{)}^{\gamma }-F_{N_{2},n})$$
where $F_{N_{2}}$ is the constant flow applied to the layer and t is the thickness of the layer. Equation (3) can be used to calculate the layer stresses needed to compensate for the through-thickness stress gradient within the film. The stress gradient is calculated from the average stress measurements taken from different wafers deposited using identical deposition process parameters and at varying AlScN thicknesses. Equation (3) is then used to interpolate between the experimental measurements of films with different thicknesses to find the through-thickness stress gradient.

## 3. Experimentation Details

The average stress through the thickness of the film is evaluated using the methods described in Section 2.1. The slope of the stress versus $F_{N_{2}}$ curve, α, is determined using a linear fit to the measured stress data. The local stress of each individual layer required to compensate for the through-thickness stress gradient is calculated using Equation (3). Cantilevers with a width of 50 µm and a length of 200 µm are realized in 500 nm thick Al_0.68_Sc_0.32_N films with out-of-plane displacements measured from the difference between the anchor and beam tip heights. The Al_1−x_Sc_x_N cantilever fabrication is shown in Figure 1 and begins with deposition of Al_0.68_Sc_0.32_N on 100 mm p-type (100) Si wafers in an Evatec CLUSTERLINE^®^ 200 II Physical Vapor Deposition System in steps (a) and (b). Table 1 summarizes the DC reactive co-sputtering parameters used to deposit the Al_0.68_Sc_0.32_N films. The Al_1-x_Sc_x_N films are deposited on a 15 nm thick AlN seed layer and a 35 nm thick gradient seed layer (Al_1→0.68_Sc_0→0.32_N) where the Sc alloying ratio is linearly varied through the thickness from 0 to 32%. This seed and gradient layer was previously demonstrated to suppress anomalously oriented grains while maintaining crystal quality [3]. The crystal quality characterized in a previous study using the full width half maximum (FWHM) of a rocking curve omega-scan showed a FWHM of 2.18° with the seed layer and 2.23° without the seed layer for films of 500 nm total (film plus seed) thickness. In step (c) Plasma Enhanced Chemical Vapor Deposition (PECVD) Silicon Nitride (SiN) is deposited and patterned using CF_4_ reactive ion etching (RIE) to form a hard mask for Al_1−x_Sc_x_N etching. In step (d) aqueous Potassium Hydroxide (KOH in 45% H_2_O) at 45 °C for 100 s is used to etch the Al_1−x_Sc_x_N and define the cantilever dimensions with SiN protecting the Al_1−x_Sc_x_N film where etching is undesired. Finally, in step (e) the SiN hard mask is stripped using CF_4_ RIE and the Al_1−x_Sc_x_N cantilevers are released from the substrate using isotropic XeF_2_ dry etching. After release, the cantilever out-of-plane deflection is measured using a VHX-5000 Digital Microscope Multiscan.

## 4. Results and Discussion

### 4.1. Through-Thickness Stress Gradients in Sputtered Al_1−x_Sc_x_N films

During film growth, stresses due to lattice mismatch, intrinsic strains and microstructure produces tensile and compressive stresses. The average stress of a sputtered Al_1−x_Sc_x_N film is strongly correlated to the value of the chamber pressure during deposition [3]. At 25 sccm process gas flow, the sputtering chamber will be at a near constant pressure of 1.09 × 10^−3^ mbar. At 25 sccm pure N_2_ flow, when Al_0.68_Sc_0.32_N is deposited to a final thickness of 500 nm, the average stress within the film will be approximately 137 MPa. The average film stress vs. thickness for Al_0.68_Sc_0.32_N films deposited under 25 sccm pure N_2_ flow using the process conditions in Table 1 is provided in Figure 2a. The average film stress is a strong function of the final film thickness due to the through-thickness stress gradient of the films. The through-thickness stress gradient can be modeled using Equation (2) in conjunction with the data in in Figure 2a where α is 41.12, β is 6.6522, and γ is 0.2194. α is determined using a linear fit of the measured average stress versus flow data shown in Figure 2b. In Figure 2a, the stress starts highly compressive and becomes more tensile as the thickness of the film increases. At lower film thicknesses, the microstructure of the film continuously changes and the grain size increases with increasing film thickness. As the thickness increases, the columnar growth of the film is more stable causing the through-thickness stress gradient to reduce and the average film stress to asymptote towards a constant value with further increases in thickness. These trends are clearly observable in Figure 2a.

### 4.2. Out-of-Plane Cantilever Deflection in Uncompensated Al_1−x_Sc_x_N Materials

Low average stress (membranes) and low through-thickness stress gradient (cantilevers) are critical for realizing high yield MEMS structures with low out-of-plane bending. The multilayer stress and stress gradient compensation approach reported in this paper can be utilized to simultaneously achieve low average stress and low through-thickness stress gradient. Table 2 provides a summary of the various flow conditions and the resulting stress and out-of-plane cantilever deflections. While low average stress is achievable in all films, only a multi-layer Al_1__−__x_Sc_x_N achieves the low through-thickness stress gradient required to realize cantilevers with low out-of-plane bending. The compressive-to-tensile stress gradient through the Al_1__−__x_Sc_x_N film thickness results in a high degree of out-of-plane bending in uncompensated Al_1__−__x_Sc_x_N films. A *Z*-axis microscope is used to measure the tip deflection with the anchor as the z = 0 reference. Figure 3 provides SEM images of the high out-of-plane bending in Al_0.68_Sc_0.32_N cantilevers deposited with a constant flow of 25 sccm.

Figure 4 shows top view SEM images of cantilevers fabricated from Al_0.68_Sc_0.32_N films from across a 100 mm wafer demonstrating that the residual stress gradient within the film not only induces bending but also generates twisting and rotations within released structures. The center of the wafer produces structures with minimal twisting while the edge of the wafer produces significant twisting depending on the location of the die. The twisting is due to the interaction of the through-thickness stress gradient with the radial variation of the average stress across the 100 mm diameter wafer and is consistent with previous studies [3] which exhibit more compressive average stresses at the wafer edge and more tensile average stresses near the wafer center. If no stress compensation is established, depending on the performance requirements for a MEMS device, the location of the die on the wafer and the orientation of the released structures can lead to differences in out-of-plane bending and twisting.

### 4.3. Stress Gradient Compensated Al_1−x_Sc_x_N Films and Cantilevers

The through-thickness stress gradient is the primary source of out-of-plane bending in released Al_1−x_Sc_x_N structures. To find the additional flow needed to compensate the stress gradient, Equation (3) is used to estimate the local stress. At 25 sccm N_2_ flow, the local stress within the first 15 nm after the seed layer has been deposited is approximated to be −424 MPa using Equation (3). At 500 nm, the local stress is 276 MPa. A compensating 424 MPa in the initial layers, −276 MPa at the top of the film, and the appropriate opposing stress gradient is required to compensate for the through-thickness stress gradient. Using Figure 2b, at 20 and 30 sccm N_2_ flow, a 500 nm Al_1−x_Sc_x_N film will yield an average stress of −450 and 317 MPa, respectively. We utilized Equation (3) to design a 2-layer stack deposited at 30 sccm (lower) and 20 sccm (upper) to compensate for the through-thickness stress gradient. Figure 5 provides an SEM image of a 2-layer Al_1−x_Sc_x_N material where the N_2_ flow is varied between layers to suppress the stress gradient and the resulting out-of-plane bending in cantilevers. Here, 30 sccm N_2_ flow is utilized during the deposition of the AlN seed, Al_1−x_Sc_x_N gradient layer, and the lower 225 nm of the bulk film while 20 sccm N_2_ flow is utilized when depositing the upper 225 nm of the film stack. The approach successfully compensates for the through-thickness stress gradient and reduces the out-of-plane cantilever bending in the center of the wafer from 109 µm for the uncompensated materials to less than 3 µm for cantilevers realized in the stress gradient compensated 2-layer material.

Figure 6b displays a 5-layer Al_0.68_Sc_0.32_N stack used to compensate for the through-thickness stress gradient. Since lower flows produce more compressive films while higher flows produce more tensile films, a layer stack is utilized where for each consecutive 100 nm layer, an additional 2.5 sccm of flow provides a tensile-to-compressive transition to cancel the original compressive-to-tensile stress gradient through the thickness. For the 5-layer material the N_2_ flow is varied over the range from 30 to 20 sccm to yield a low average stress in addition to a low through-thickness stress gradient. After release, the cantilevers remain consistently flat, as shown in Figure 6a and Figure 7, especially when compared to the cantilevers formed in the uncompensated films. The cantilevers in Figure 4, Figure 5, Figure 6 and Figure 7 use the same naming conventions depicted in Figure 3a,b. In Figure 6a the maximum tip deflection is approximately 5.8 +/− 0.4 µm, −7.6 +/− 0.4 µm, and −4.0 +/− 0.4 µm when measured from the 1, 2, and 3 positions, respectively. The 1, 2, and 3 structures exhibited the same behavior and deflection as those directly across from them, namely cantilevers 5, 6, and 7. Table 2 compares the average wafer stress and cantilever tip deflection for the uncompensated and stress gradient compensated Al_0.68_Sc_0.32_N materials. The out-of-plane cantilever displacement for the 5-layer, low stress material at position 2 is reduced by more than 14-fold for the center die and 19-fold for the east die 35 mm from the wafer center. Overall, substantial reductions in out-of-plane tip displacement are observed for all cantilevers fabricated in the stress gradient compensated films. The 5-layer film does not have an AlN/gradient seed layer and possess a higher tip deflection than the 2-layer film with seed layer. This is because the gradient in the N_2_ process gas flow was designed using the data from Figure 2 where all the films have the seed layer. Table 3 confirms the precise control of the tip deflection by varying the range of N_2_ gas flow for a 2-layer material stack. The range of gas flows controls the stress gradient while the mean N_2_ gas flow controls the average stress within the film. In Table 3, a 2.5 sccm increase of the range from 27.5–22.5 sccm to 30–22.5 sccm reduces the deflection 2-fold.

### 4.4. Discussion of Stress Gradient Cancellation Trends

This work provides, for the first-time, methods to individually control the stress and stress gradient in Al_1−x_Sc_x_N films while maintaining film quality. Figure 8 shows the out-of-plane displacement along the length of the cantilever for the uncompensated and 2-layer compensated Al_0.68_Sc_0.32_N materials while Table 3 summarizes the out-of-plane tip displacement. The compensated cantilever tip bending confirms that the radius of curvature can be controlled in the released Al_1−x_Sc_x_N structures. The multilayer gas gradient method can be utilized to simultaneously and independently control both the average stress, via the average N_2_ flow, and through-thickness stress gradient, via the through-thickness variation of the N_2_ flow, in Al_1−x_Sc_x_N thin films. Previously reported methods to control Al_1−x_Sc_x_N average film stress do not provide control of the through-thickness stress gradients within the film. While the previous RF substrate bias method reported by Knisley [14] achieved independent control of stress and through-thickness stress gradient in AlN films and demonstrated reduced radius of curvature and tip displacement in AlN cantilevers, use of an RF substrate bias is less suitable for Al_1−x_Sc_x_N. Addition of an RF substrate bias results in more compressive film stress and is a good technique for stress control of AlN where the films are highly tensile without an RF substrate bias [14]. Al_0.68_Sc_0.32_N, by contrast, is highly compressive when deposited at the low process pressures that suppress formation of anomalously oriented grains (AOGs) without using an RF substrate bias [3]. Thus, addition of an RF substrate bias to an Al_0.68_Sc_0.32_N growth will require even higher process pressures to achieve near-neutral average film stress, and under such process conditions a large number of AOGs would be expected.

## 5. Conclusions

This study reports methods to fabricate Al_0.68_Sc_0.32_N films and cantilevers with low average stress and with low through-thickness stress gradients. Al_1−x_Sc_x_N films were optimized to control stress for N_2_ flows between 20 to 30 sccm. The average stress within the films ranged from 78.6 MPa to 349.6 MPa. The out-of-plane tip deflection for 100 µm long cantilevers fabricated in 500 nm thick Al_0.68_Sc_0.32_N films was reduced from >109 µm for films without stress gradient compensation to less than 3 µm and 8 µm for 2- and 5-layer compensated film stacks for dies studied in the wafer center. The resulting deposition parameters provide methods to control stress and through-thickness stress gradients in highly Sc alloyed AlN materials and are promising for next-generation MEMS devices.

## Figures and Tables

**Figure 1 micromachines-13-01169-f001:**
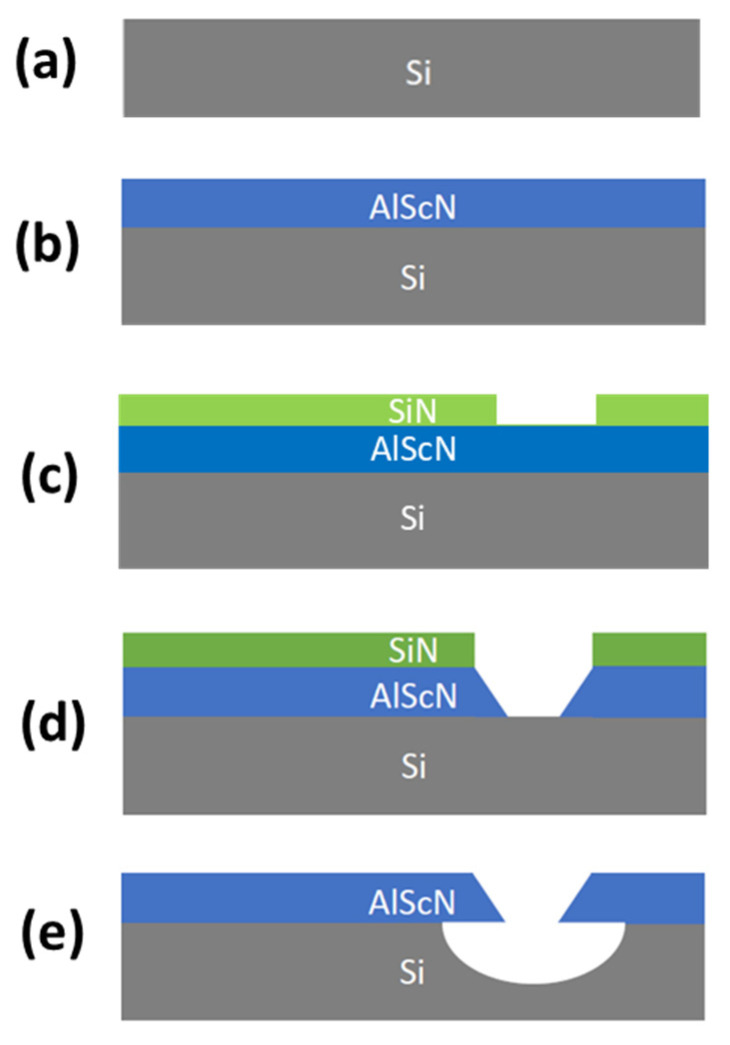
Fabrication process for realizing Al_1−x_Sc_x_N cantilevers with (**a**) p-type (100) Si wafer (**b**) Al_1−x_Sc_x_N deposition using Evatec CLUSTERLINE^®^ 200 II PVD system (**c**) PECVD SiN deposition and patterning using CF_4_ RIE (**d**) KOH in 45% H_2_O etch of Al_1−x_Sc_x_N (**e**) SiN hard mask stripped using CF_4_ RIE and Al_1−x_Sc_x_N cantilevers released using isotropic XeF_2_ dry etching.

**Figure 2 micromachines-13-01169-f002:**
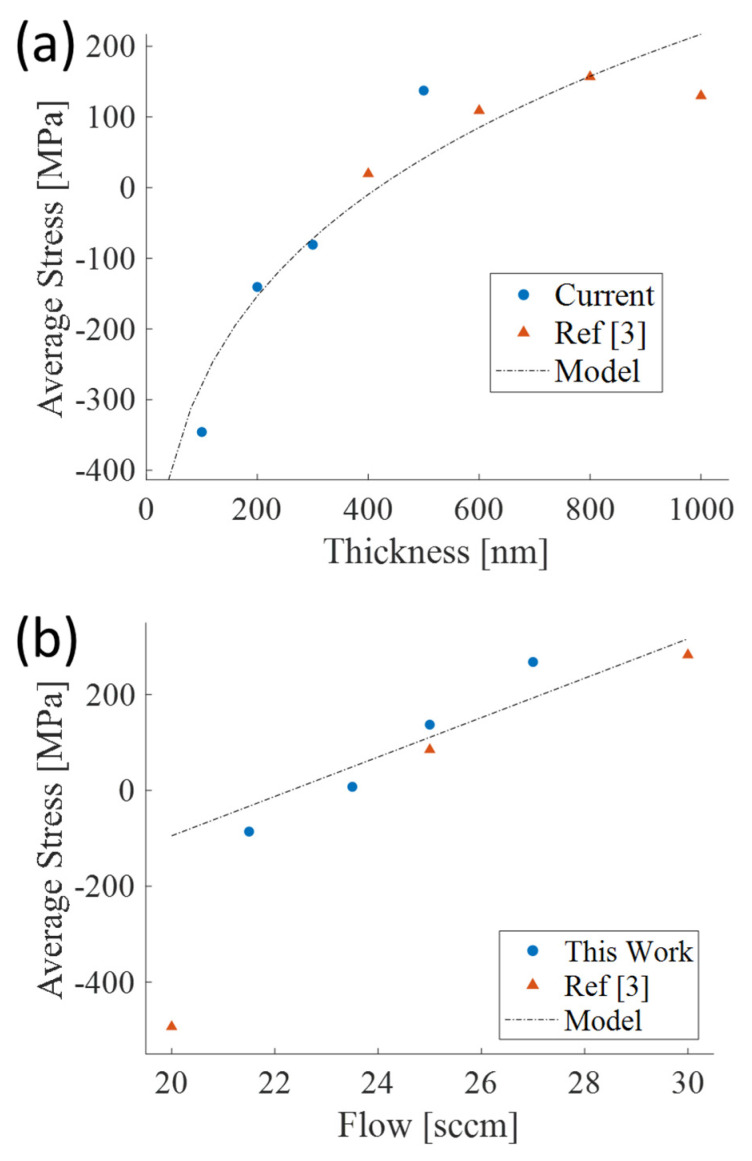
Average stress plots of PVD deposited Al_0.68_Sc_0.32_N films with (**a**) Average stress versus thickness plot at a constant 25 sccm N_2_ flow and (**b**) Average stress versus flow for 500 nm Al_0.68_Sc_0.32_N with pure N_2_ flow from 20–30 sccm [3].

**Figure 3 micromachines-13-01169-f003:**
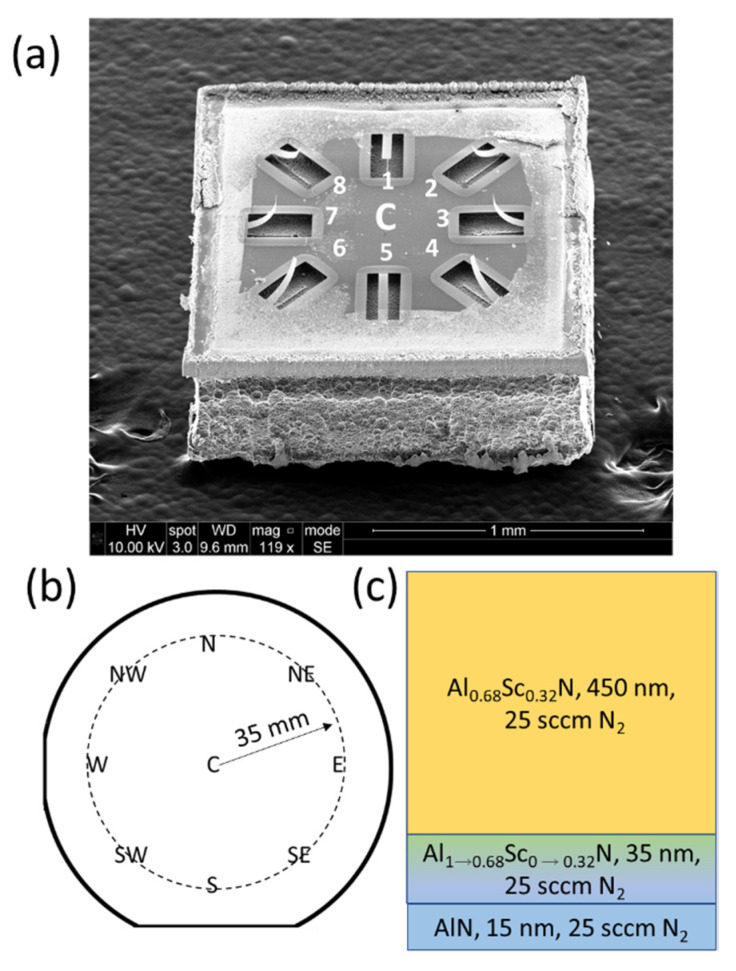
Graphics for a 500 nm PVD sputter-deposited Al_0.68_Sc_0.32_N film where the N_2_ process gas flow is held constant through the entire deposition, (**a**) 45-degree SEM image of center die (C) fabricated from a 100 mm wafer. Each die has 8 released structures which are labeled. Note the high out-of-plane deflection that is clearly visible in the uncompensated structures. (**b**) Schematic of locations on the 100 mm wafer where a die is pulled 35 mm from the center for imaging and measuring out-of-plane displacements. One die was pull from the north (N), northeast (NE), east (E), southeast (SE), south (S), southwest (SW), west (W), and northwest (NW) locations of the wafer (**c**) Stack-up of film with constant flow composed of a seed layer, gradient seed layer (Al_1→0.68_Sc_0→0.32_N) [3] and Al_0.68_Sc_0.32_N layer.

**Figure 4 micromachines-13-01169-f004:**
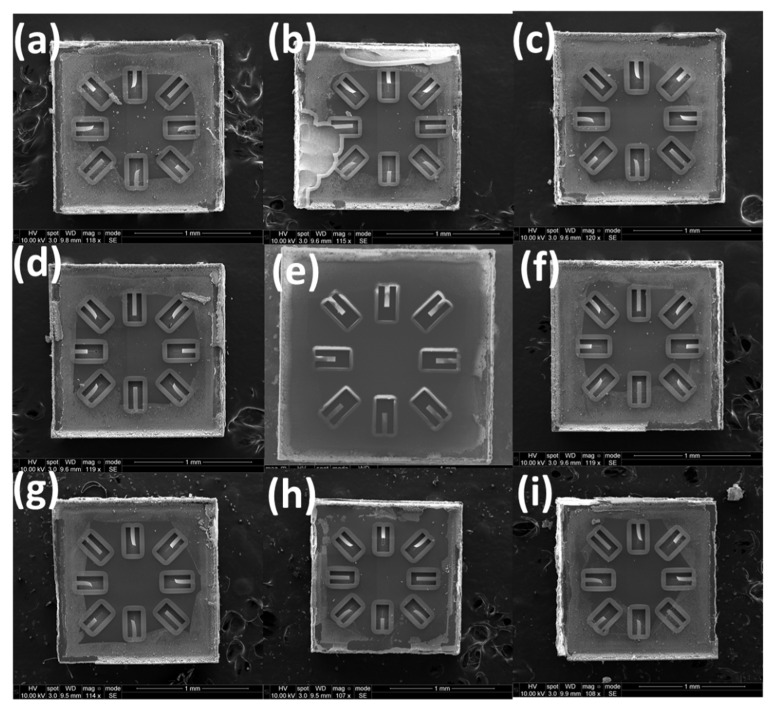
(**a**–**i**) SEM images of cantilevers formed from 500 nm thick PVD Al_0.68_Sc_0.32_N films deposited under a constant N_2_ flow of 25 sccm. Each die was pulled from the north (N), northeast (NE), east (E), southeast (SE), south (S), southwest (SW), west (W), and northwest (NW) locations of the wafer shown in Figure 3b.

**Figure 5 micromachines-13-01169-f005:**
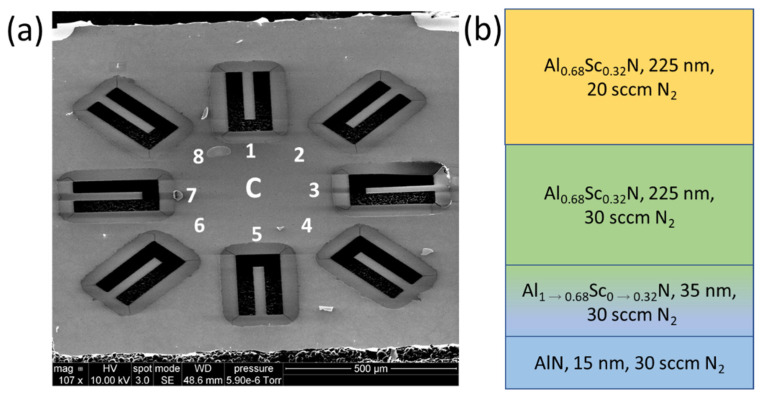
Graphics for multi-layer 500 nm PVD sputter-deposited Al_0.68_Sc_0.32_N film where the N_2_ process gas flow is changed between two layers with 30 sccm utilized on the bottom and 25 sccm on the top layer, (**a**) 45-degree SEM image of the center die (C) fabricated from a 100 mm wafer. Each die has 8 released structures. (**b**) Stack-up of 2-layer film composed of a seed and gradient layer (Al_1→0.68_Sc_0→0.32_N) [3] to suppress AOGs and two equal thickness layers with different N_2_ process gas flows designed to compensate for the native through-thickness stress gradient.

**Figure 6 micromachines-13-01169-f006:**
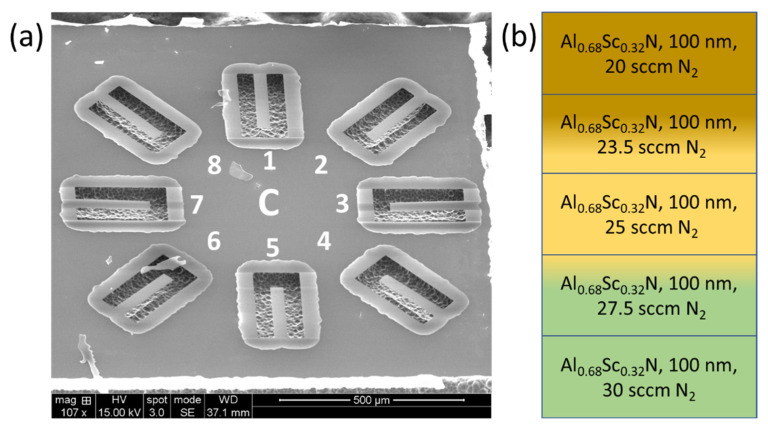
Graphics for multi-layer 500 nm PVD sputter-deposited Al_0.68_Sc_0.32_N film, (**a**) 45-degree SEM image of the center die (C) fabricated from a 100 mm wafer. (**b**) Stack-up with five equal thickness layers with different N_2_ process gas flows to compensate for the through-thickness stress gradient.

**Figure 7 micromachines-13-01169-f007:**
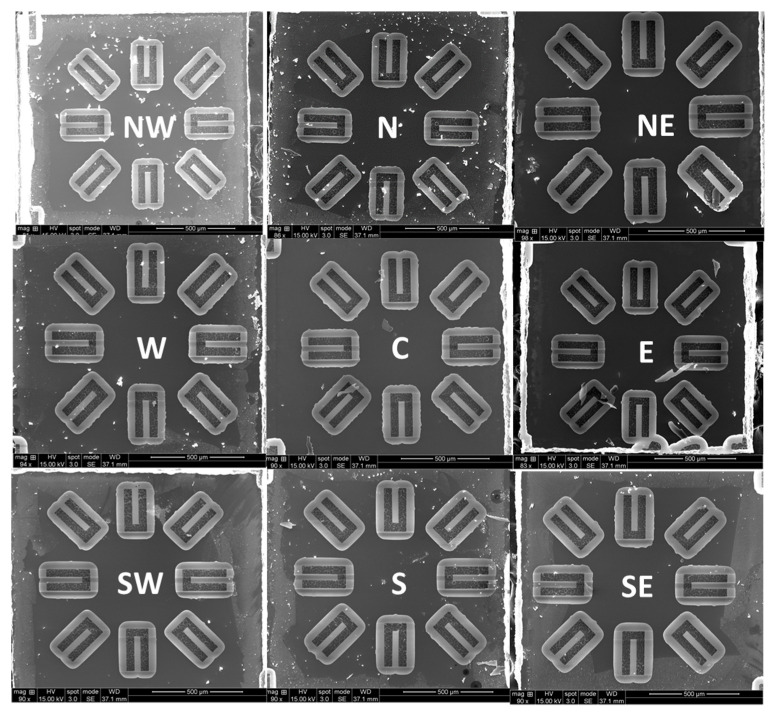
Top view SEM images of cantilevers formed from a 5-layer, 500 nm total thickness, PVD deposited Al_0.68_Sc_0.32_N where the process gas flow for each layer is linearly changed from 30 to 20 sccm through the five layers. Each die was pull from the north (N), northeast (NE), east (E), southeast (SE), south (S), southwest (SW), west (W), and northwest (NW) locations of the wafer shown in Figure 3b.

**Figure 8 micromachines-13-01169-f008:**
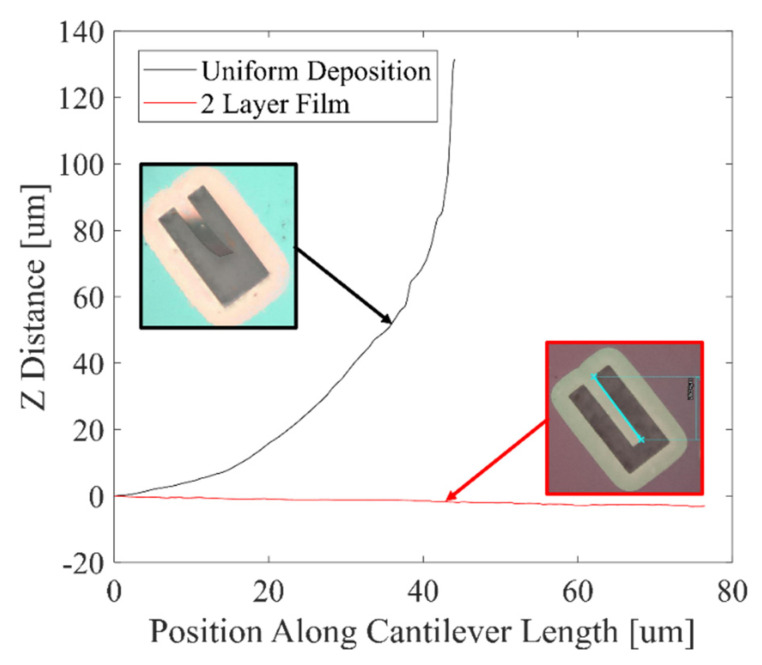
Z deflection for 500 nm PVD sputter-deposited Al_0.68_Sc_0.32_N films where the N_2_ process gas flow was applied at constant flow throughout the deposition (black) and a 2-layer 30/20 sccm N_2_ flow stack (red).

**Table 1 micromachines-13-01169-t001:** Summary of sputter deposition parameters for Al_0.68_Sc_0.32_N.

Process Parameter	Value
Temperature	350 °C
Sputter Power Al Cathode	1000 W
Sputter Power Sc Cathode	555 W
DC Pulsing Frequency	150 kHz
N_2_ Flow	20–30 sccm
Film Thickness	100–1000 nm
Base Pressure	<3 × 10^−7^ mbar

**Table 2 micromachines-13-01169-t002:** Cantilever tip deflection for 500 nm Al_0.68_Sc_0.32_N with and without stress gradient compensation.

		# of Layers		
		Single	Double	Quintuple
N_2_ Flow [sccm]		25	30/20	30/27.5/25/23.5/20
Average Stress [MPa]		137.4	349.6	78.6
Seed Layer		Yes	Yes	No
**Wafer Location**	**Device Position**	**Out-of-Plane Deflection [µm]**		
Center	1	115.2 +/− 1.2	−3.2 +/− 0.1	−5.8 +/− 0.4
	2	108.6 +/− 4.2	−0.7 +/−0.2	−7.6 +/− 0.4
	3	117.1 +/− 0.9	0.0 +/− 0.1	−4.0 +/−0.4
North	1	50.1 +/− 0.1	0.9 +/− 0.3	7.4 +/− 0.4
	2	121.0 +/− 0.7	−1.4 +/− 0.2	−7.7 +/− 0.3
	3	145.9 +/− 0.2	−2.8 +/− 0.8	−20.0 +/− 0.1
Northeast	1	125.1 +/− 0.3	−7.0 +/− 0.2	−15.6 +/− 0.4
	2	49.3 +/− 0.1	−4.0 +/− 0.1	−21.1 +/− 0.6
	3	120.6+/− 0.4	0.1 +/− 0.1	−0.9 +/− 0.3
East	1	128.9 +/− 0.9	0.9 +/− 0.2	−20.9 +/− 1.9
	2	126.9 +/− 42	6.2 +/− 0.2	6.4 +/− 0.2
	3	60.3 +/− 0.6	6.6 +/− 0.1	22.0 +/− 0.2

**Table 3 micromachines-13-01169-t003:** Cantilever tip deflection for a die near the center of a 500 nm Al_0.68_Sc_0.32_N film with AlN and Sc gradient (Al_1→0.68_Sc_0→0.32_N) seed layers.

N_2_ Flow [sccm]	25	27.5/22.5	30/22.5	30/20
Seed Layer	Yes	Yes	Yes	Yes
**Device** **Position**	**Out-of-Plane Deflection [µm]**			
1	115.2 +/− 1.2	60.8 +/− 0.1	24.1 +/− 0.1	−3.2 +/− 0.1
2	108.6 +/− 4.2	55.0 +/− 0.1	28.2 +/− 1.2	−0.7 +/− 0.2
3	117.1 +/− 0.9	51.4 +/− 0.3	30.5 +/− 0.4	−0.0 +/−0.1

## Data Availability

The data that support the findings of this study are available from the corresponding author upon reasonable request.

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
