# Peer review of "Compensation of the Stress Gradient in Physical Vapor Deposited Al1−xScxN Films for Microelectromechanical Systems with Low Out-of-Plane Bending"

_micromachines, 2022, doi:10.3390/mi13081169_

Round 1

Reviewer 1 Report

The authors of the manuscript entitled “Compensation of the Stress Gradient in Physical Vapor Deposited AlScN films for Microelectromechanical Systems with Low Out-of-Plane Bending” are studying the average stress and the stress gradient in the sputter deposited AlScN films deposited on silicon. They demonstrate that process gas flow variation enables to change the stress state of the film. This method enables to reduce the out-of-plane bending of the final cantilever structure.  

All in all, this manuscript shows interesting results that are worth sharing with the scientific community. However, it cannot be published as it is. Literature review is not sufficient, and manuscript misses a real discussion part. Please find my comments and questions to clarify some points of the manuscript.

1.       Literature about work on stress compensation of thin film structures in MEMS is not covered in the introduction. Please add corresponding references.  

2.       There is an error on the 8th and 9th line under the Table 1.

“Error! Ref[1]erence source not found.1 and begins”

3.       Please explain how do you measure the stress gradient through the thickness of the film?

4.       Could you provide another measurement technique results, for example sin2(ψ) method in XRD, and confirm the full-sheet film stress level with this technique?

5.       The font of the text under Table 3 doesn’t match the font of the rest of the text in the manuscript.

6.       Discussion part is very short in this manuscript. Please discuss the results of your work in comparison to results published on the same topic. Why is your approach interesting and innovative?  

Reviewer 2 Report

The authors demonstrate the importance of controlling the average film stress and especially the stress gradient within the films to reduce bending and torque of freestanding AlScN cantilevers for MEMS applications. The study is scientifically sound and of high interest to the AlScN community.

Before publication, some minor polishing should be done.

- The importance of the AlScN material for future MEMS devices and its possible applications should be briefly described in the introduction.

- Please shortly motivate and decribe the deposition of the Sc-gradient seedlayer geometry in the experimental part. Is it to reduce stress from the beginning or to improve crystal quality? Did the authors characterize the crystalline quality of all deposited films by XRD?

- The authors may want to share any specifics about their sputter tool

- The quality of all sketches and plots could be improved (low resolution)

- Please check chemical formulas and the use of subscript where appropriate, e.g. N2 and try to stay consistent in the manuscript, Al0.68Sc0.32N / Al68Sc32N / Al100-68Sc0-32N / AlScN (where you mean a specific concentration). My suggestion is to use the chemically correct nomenclature of Al1-xScxN and Al0.68Sc0.32N

- Please use the chance to eliminate typos, e.g. p.5 "though" thickness stress gradients, p1. AlN Aluminum Nitride (AlN), p7. "such as stack", p7. "output-of-plane cantilever bending"

- the referencing to Figure 1 is corrupt "Error! Reference source not found.1"

- please give reference to your prior work [3] after the sentence "We previously demonstrated that by controlling N2 flow between 20-30 sccm that 500 nm thick Al68Sc32N materials, free of AOGs, with low surface roughness and strong c-axis orientation, could be sputter deposited on Si with a controlled average film stress ranging from -458 to 287 MPa."

- in section II A, the paramters R0 and R are introduced twice

- How do the authors explain the less bending in double layers in comparison to 5-Layer geometry although the average stress is higher for the 2-layer systems?

Reviewer 3 Report

I read with interest the manuscript titled “Compensation of the Stress Gradient in Physical Vapor Deposited AlScN films for Microelectromechanical Systems with Low Out-of-Plane Bending” in which the authors report on how to diminish the stress gradient in AlScN thin films through multilayer-deposition of the films by changing the N­2 gas flow.

The work reported in the manuscript is new and is of high interest for piezo-MEMS scientific and engineering community. However, the manuscript itself is not very clearly written and its structure should be improved before its publication. My main comments and suggestions are listed below.

Page 3 – Figure reference is missing.

Figure 3c – the use of a seed layer and gradient seed is not mentioned in the experimental details. Please add.

The authors do analysis about the behaviour of cantilevers across different parts of the 100 mm wafer and within each die. I have two main issues with this analysis:

-        It is difficult to make any conclusions from SEM images the authors give. More specifically, describing Figure 4, the authors give the following statement: “The edges of the wafer are more compressive while the center of the wafer is more tensile. The center of the wafer produces structures with minimal torque while the edge of the wafer produces high or low rotations depending on the location of the die.” From Figure 4 as shown in the manuscript it is impossible to make this conclusion. Please either improve its quality, present different type of analysis or omit this statement.

-        What is conclusion of the analysis at different parts of the wafer?

Table 3. Each die had 8 cantilevers, only 3 are presented. How were they chosen?

Table 3. Variation of deflection within a single die seems to be larger than the variation at different parts of the wafer. Where is this variation coming from? Please discuss.

Table 3. It seems that double-layer approach works better than quintuple-layer (deflection data). Why? Can the authors discuss this in the manuscript?

Finally, I would strongly suggest including in the manuscript some electrical characterization. For instance, does lower stress gradient have influence on piezoelectric (or even ferroelectric) properties of AlScN films?

Round 2

Reviewer 1 Report

The authors made changes in the manuscript. The manuscript can be published now.

Reviewer 3 Report

The authors adequately addressed my questions/comments and I agree with publication of the manuscript.